# Prostaglandin E2 (PGE2) and Roflumilast Involvement in IPF Progression

**DOI:** 10.3390/ijms241512393

**Published:** 2023-08-03

**Authors:** Noa Moshkovitz, Gali Epstein Shochet, David Shitrit

**Affiliations:** 1Pulmonary Department, Meir Medical Center, Kfar Saba 44281, Israel; noamosh@gmail.com (N.M.); galieps@gmail.com (G.E.S.); 2Sackler Faculty of Medicine, Tel Aviv University, Tel Aviv 69978, Israel

**Keywords:** IPF, PGE2, Roflumilast, extracellular matrix, fibroblast

## Abstract

The ECM propagates processes in idiopathic pulmonary fibrosis (IPF), leading to progressive lung scarring. We established an IPF-conditioned matrix (IPF-CM) system as a platform for testing drug candidates. Here, we tested the involvement of a PGE2 and PDE4 inhibitor, Roflumilast, in the IPF-CM system. Primary normal/IPF tissue-derived human lung fibroblasts (N/IPF-HLFs) were cultured on Matrigel and then removed to create the IPF-CM. N-HLFs were exposed to the IPF-CM/N-CM with/without PGE2 (1 nM) and Roflumilast (1 µM) for 24 h. The effect of the IPF-CM on cell phenotype and pro-fibrotic gene expression was tested. In addition, electronic records of 107 patients with up to 15-year follow-up were retrospectively reviewed. Patients were defined as slow/rapid progressors using forced vital capacity (FVC) annual decline. Medication exposure was examined. N-HLFs cultured on IPF-CM were arranged in large aggregates as a result of increased proliferation, migration and differentiation. A PGE2 and Roflumilast combination blocked the large aggregate formation induced by the IPF-CM (*p* < 0.001) as well as cell migration, proliferation, and pro-fibrotic gene expression. A review of patient records showed that significantly more slow-progressing patients were exposed to NSAIDs (*p* = 0.003). PGE2/PDE4 signaling may be involved in IPF progression. These findings should be further studied.

## 1. Introduction

Idiopathic pulmonary fibrosis (IPF) is a progressive interstitial pneumonia that is characterized by scar tissue formation and excessive extracellular matrix (ECM) deposition, resulting in the loss of lung function [1,2]. A major therapeutic challenge arises from the variability in the clinical course, as IPF patients may exhibit distinct, and unpredictable, patterns of disease progression [3,4,5]. Some patients show a slowly progressive disease (‘slow’ progressors (SP)), often with long duration of symptoms before diagnosis, while others display a rapidly progressive course (‘rapid’ progressors (RP)) [4]. Disease course may also be complicated by acute exacerbations, with dismal prognosis [3,6]. Although there are some distinguishing markers between the groups [3,7,8], most clinical trials do not take them into consideration [9,10,11]. At present, Nintedanib and Pirfenidone are the only FDA-approved IPF treatments [12,13]. Nonetheless, IPF still carries poor prognosis, as these treatments are only able to slow down disease progression. Thus, additional targets and treatment options are needed.

The ECM plays a significant role in fibrosis progression. As reviewed [14], IPF-ECM alone can induce normal lung fibroblasts to turn into activated myofibroblasts. Once formed, IPF-ECM sets up a pro-fibrotic feedback loop that is capable of sustaining progressive fibrosis. In support of this hypothesis, our group developed an in vitro IPF conditioned matrix (IPF-CM) system, which utilizes human primary lung fibroblasts (HLF) derived from patients with IPF. Normal tissue-derived HLFs (N-HLFs) added to IPF-CM were shown to have increased migration and elevated pro-fibrotic markers. This platform allows us to investigate drug function with regard to fibroblast–ECM interaction [15,16,17].

COX-2 expression is regulated via several signals of injury and inflammation [18]. An abnormal expression of COX-2 has been linked to cancer and chronic inflammatory pathologies, including fibrosis [19]. Roflumilast, a long-acting selective inhibitor of the phosphodiesterase-4 (PDE4) enzyme, was shown to have anti-inflammatory effects in lung fibroblast models, and it is prescribed for other conditions, such as chronic obstructive pulmonary disease (COPD) [20,21]. PDE4 inhibition affects a broad spectrum of lung fibroblast functions, such as myofibroblast transition and ECM generation in vitro and mitigates bleomycin-induced lung fibrosis in vivo [22,23]. PDE4 inhibition was recently reviewed by Schick and Schlegel [24] as a possible therapeutic target.

In this study, we explored the effect of PGE2 and PDE4 inhibition on the IPF-CM system as further support to previously published evidence for this approach in IPF.

## 2. Results

### 2.1. PGE2 in Combination with Roflumilast Inhibits IPF-CMs’ Pro-Fibrotic Effects on N-HLFs

The main outcome of the IPF-CM system is large aggregate formation [15]. As previously described, the IPF-CM elevates the aggregate size in comparison to N-CM after 24 h (*p* < 0.001) as well as the total number of cells. Recently, we combined the aggregate size and aggregate number values to create a new, more accurate parameter, the average aggregate size, as published in [17]. Here, we tested whether the PGE2 with/without the PDE4 inhibitor, Roflumilast, will have an effect in this system. The addition of PGE2 or Roflumilast alone showed a partial effect on relative aggregate size (IPF-CM vs. N-CM, Figure 1A) and cell counts (Figure 1B). However, the addition of PGE2 and Roflumilast resulted in a complete inhibition of large aggregate formation as well as a reduction in the relatively elevated cell counts (*p* < 0.0001 and *p* < 0.05, respectively, Figure 1A,B). To test whether this was a result of reduced cell migration or increased cell death, we performed a wound-healing assay and tested cell death by flow cytometry with PGE2 with/without Roflumilast. Results show that the addition of PGE2 and Roflumilast inhibits HLF migration (*p* < 0.05, Figure 1C) and total cell death (*p* < 0.05, Figure 1D). Thus, the combination of PGE2 and Roflumilast leads to a complete inhibition of IPF-CMs’ effects on N-HLFs.

Pro-fibrotic gene expression was also tested. We found that COL1A, ACTA2, TGFB1 and MMP2, which were not affected by PGE2 alone, were completely inhibited by the addition of PGE2 + Roflumilast (*p* < 0.05, Figure 2A–D). IL-8 expression was not affected by the addition of PGE2 or Roflumilast (Figure 2E). Interestingly, HIF1A expression was completely inhibited by PGE2 alone, and it showed a similar result with the addition of Roflumilast (*p* < 0.05, Figure 2F). These findings may suggest that PGE2 pathway inhibition, using NSAIDs for instance, may inhibit disease progression.

In order to test the direct effect of PGE2 and Roflumilast on HLFs, cells were exposed to PGE2 and TGFβ with/without Roflumilast for 24 h. In accordance with the IPF-CMs’ results, the addition of TGFβ increased, while PGE2 significantly decreased ACTA2 and COL1A mRNA expression. The addition of Roflumilast resulted in partial inhibition (Figure 3A,B). To assure this was not a result of reduced cell viability, cell death and numbers were also tested (Figure 3C,D). In fact, PGE2 addition to HLFs reduced cell death and slightly increased cell counts, suggesting that PGE2 addition promoted cell viability. Moreover, the addition of PGE2 together with TGFβ and Roflumilast did not change the rate of cell death or cell counts, indicating that the changes in pro-fibrotic gene expression were not directly associated with reduced cell viability in the IPF-CM system. Our previous work [17] showed that the HIF1A pathway is significantly activated in the IPF-CM using RNAseq and that the plasminogen activator inhibitor-1 (PAI1) protein (SERPINE1) is elevated. Thus, we tested PAI1 expression in the HLFs and found that exposure to PGE2 significantly reduced PAI1 protein expression. HIF1A protein levels were increased following TGFβ addition, yet they were not significantly changed by PGE2/Roflumilast directly (Figure 3E).

### 2.2. Slow-Progressing IPF Patients Were More Exposed to NSAIDs

To retrospectively test this hypothesis, we classified a cohort of IPF patients into slow and rapid progressors (SP/RP) and reviewed their medication intake. Study population included 107 patients diagnosed with IPF; of them, 62 were classified as SP and 45 as RP, as described in the Section 4. The average age at diagnosis was 66 ± 11 with 67% males. Patient demographics and comorbidities are presented in Table 1. Similar to previous reports [3], male predominance was significantly more pronounced in the RP group, which also included more patients with renal failure, yet it had no patients with malignancy (Table 1). Overall, survival, as well as transplant-free survival, was significantly higher in the SP group (*p* < 0.001, Figure 4).

Moreover, during follow-up, pulmonary function tests were significantly better in patients from the SP group, including forced vital capacity (FVC), total lung capacity (TLC), DLCO and 6 min walking test (6MWT) parameters (Table 2).

Most patients with IPF, as with other patients over the age of 60, suffer from various comorbidities and are prescribed several concomitant medications. These include proton pump inhibitors, anticoagulants, statins, antihypertensive drugs, nonsteroidal anti-inflammatory drugs (NSAIDs), etc. [25]. Thus, we examined medication exposure to the five major groups of concomitant treatments between the SP and RP patients during the follow-up period. There were no significant differences between the groups regarding usage of statins (*p* = 0.833, Figure 5A), angiotensin-converting enzyme (ACE) inhibitors (*p* = 0.71, Figure 5B), beta blockers (*p* = 0. 409, Figure 5C) or serotonin-specific reuptake inhibitors (SSRI) (*p* = 0.271, Figure 5D). Interestingly, there was a significant difference in exposure to NSAIDs, with 15 SP patients being exposed to this class of drugs as opposed to only two patients in the RP group (*p* = 0.003, Figure 5E).

## 3. Discussion

The ECM plays a major role in IPF progression [15]. In this work, we utilized the already established IPF-CM system to explore the involvement of PGE2 in the IPF–HLF–ECM interplay. We showed that PGE2 and Roflumilast combination completely inhibited IPF-CM pro-fibrotic effects on N-HLFs. These findings were then supported by a small cohort of patients, which were followed up for up to 15 years, showing that in fact, slow-progressing IPF patients were more exposed to NSAIDs than the rapidly progressing group.

PGE2 was shown to inhibit fibroblast proliferation, collagen synthesis and the modulation of TGFβ-induced fibroblast to myofibroblast transition [26,27,28]. Moreover, increased PGE2 levels were reported in bleomycin-induced lung fibrosis in murine models [29]. The crosstalk between TGFβ, which was previously found to be activated in the IPF-CM system [30], to PGE2 is complex [31,32,33,34]. On the one hand, PGE2 can inhibit fibroblast proliferation, the synthesis of collagen and modulation of TGFβ-induced fibroblast to myofibroblast transition [19]. On the other hand, TGFβ is also known to increase PGE2 levels. This issue was reviewed by Bozyk et al. [19], where they suggested a way by which PGE2 can either promote or inhibit the fibrotic process, according to the four E Prostanoid (EP) receptors (EP1–EP4), while the cellular response to prostaglandins depends on the receptor they harbor and the PGE2 concentration [35,36]. Since these were not tested, it is hard to draw further conclusions. Moreover, matrix stiffness also affects prostaglandin expression [37]. Here, we showed that the addition of PGE2 alone was not sufficient, and the combination of several factors is required for complete inhibition.

Roflumilast, a long-acting selective inhibitor of PDE4, had a positive effect by reducing aggregate size and cell migration. The Roflumilast + PGE2 combination inhibited the increase in pro-fibrotic gene expression. This is supported by in vitro [22,23], as well as by in vivo [38] works that showed PDE4 inhibition to reduce lung fibrosis. Our results demonstrate these findings in a human-derived model of primary lung cells. This is important, as the UIP pattern is specific to humans [39], thus giving further validation to previously published results.

Interestingly, Roflumilast with PGE2 had a significantly stronger effect than Roflumilast alone. PDE4 inhibitors, such as Roflumilast, prevent the breakdown of cAMP, while TGFβ modulates cAMP by altering the metabolism of PGE2. The effect of PDE4 inhibitors may have been mediated through a cAMP-stimulated protein kinase, and it depended on fibroblast production of PGE2 and TGFβ-induced PGE2 production. From our results, which show a significant effect of PGE2 addition alone, we can carefully assume that PGE2 levels in the culture are not saturated. Thus, by adding PGE2, we somewhat increased the ‘inflammatory’ state of the culture, also affecting TGFβ signaling and mimicking an inflammation that often, but not always, is present in IPF. Cortijo et al. [23] suggested several mechanisms by which Roflumilast may attenuate lung fibrosis. Of them, it was shown that Roflumilast reduced collagen mRNA expression as well as TGFβ1 formation in BLM treated mice. Furthermore, an in vitro study using fetal lung fibroblasts stimulated with TNF-alpha highlighted that the effects of Roflumilast on intercellular adhesion molecule-1 (ICAM-1) and eotaxin release were more prominent in the presence of PGE2, as in our system. Moreover, Sabatini et al. [40] pointed out that Roflumilast N-oxide diminished the TGFB1-induced expression of alpha-SMA and transcripts of connective tissue growth factor (CTGF) and fibronectin in the presence of basic fibroblast growth factor (bFGF). This is interesting, as our previous reports showed increased levels of both bFGF and TNF-alpha in the supernatants of IPF HLF and IPF-CM cultures [41], and that nintedanib, which blocks the FGF receptor (FGFR), was shown to have a significant impact in the IPF-CM system [15,16]. Additional reports suggested that PDE4 inhibitors, together with TGFβ, resulted in augmented PGE2 production along with an increased expression of COX mRNA and protein [21].

As recently reviewed by Claire Lugnier, PDE4 inhibition has many possible downstream pathways, leading to reduced inflammation, reduced oxidative stress, TNF-α, and cytokine production [42], which are all relevant for IPF progression.

Our findings require further research, as there may be several mechanisms of action to achieve the reduction in aggregate formation, which was previously shown to be correlated with the pro-fibrotic phenotype of the HLFs. However, although we did not elucidate the exact mechanism, we prove further evidence of the potential benefit of this already approved treatment for another indication. Due to the inclusion criteria, patients with COPD were not included, and thus, no data regarding Roflumilast were available in this cohort.

HIF1A is a facilitator of fibrosis, and it has been considered a pro-inflammatory factor [43] as well as a regulator of pro-fibrotic mediator production [44,45]. An interesting finding was that the HIF1A was significantly inhibited by PGE2 itself. A recent study conducted in our lab showed that HLFs cultured on IPF-CM showed an over-expression of HIF1A. Moreover, the HIF1A signaling pathway was the most over-expressed pathway in the analysis of RNA-sequencing of HLFs cultured on the IPF-CM with significant involvement of PAI1 (SERPINE1) [17]. Ivanova et al. showed that PGE2 suppresses HIF1A expression, thus supporting our results [46]. That study also deducted that PGE2 could be used to reduce fibrosis in IPF. Moreover, it was recently shown that PDE4 inhibition significantly decreased CTGF, PAI-1, collagen 1A1 and fibronectin mRNA in TGFβ-stimulated human mesangial cells [47]. These results are in accordance to our results, as the IPF-CM system was previously shown to activate TGF-β signaling [15].

Despite its inevitably progressive nature, IPF is characterized by a highly variable disease course, which makes the natural history of the disease largely unpredictable in individual patients [48]. Our study aimed to examine the influence of certain common medication groups, unrelated to IPF, on disease progression. The majority of patients with IPF, as other patients over the age of 60, suffer from various comorbidities and are prescribed several concomitant medications. These include proton pump inhibitors, anticoagulants, statins, antihypertensive drugs, nonsteroidal anti-inflammatory drugs (NSAIDs), etc. [25]. Some of these treatments were already implicated in having an effect on fibrotic processes. For instance, angiotensin type 1 receptor blockers (ARB1) and ACE inhibitors were previously shown to suppress the release of TGFβ and diminish connective tissue synthesis [49]. A recent study that retrospectively explored the effect of cardiovascular drugs on IPF progression in 323 patients suggested that statin therapy may be beneficial for IPF progression [50]. Similarly, SSRIs were also implicated to have an effect of ILD progression [51].

In this relatively preliminary analysis, we examined medication exposure to five major groups of concomitant treatments between the SP and RP patients during the follow-up period of up to 15 years. The medications were grouped into major categories, as the population size was limited. Here, we demonstrated that patients classified as RP were less exposed to NSAID treatment during the course of their illness. Other treatment groups did not reach significance, although some trends were observed. It is important to emphasize, however, that drug exposure in our study was not limited to selective COX-2 inhibitors. Naproxen, a classical NSAID, was found to be effective in reducing lung inflammation and preventing collagen accumulation in a murine model of bleomycin-induced pulmonary fibrosis [52]. Interestingly, metastatic and fibrotic processes were shown to share similar characteristics as well as signaling pathways (e.g., TGFβ and MAPK) (reviewed in [53]). One of the major pathways in cancer, c-Met, was shown to be inhibited by NSAIDs when administrated at an early stage [54,55]. In fact, studies suggested that the combination of COX-2 inhibitors with current treatments may improve survival from lung cancer [56]. Aspirin not only blocks the biosynthesis of prostaglandins but also stimulates the endogenous production of anti-inflammatory and pro-resolving mediators [57,58].

Our study has several limitations. First of all, it is a retrospective cohort based on electronic databases where medication exposure is based on those purchased by patients. We were also not able to assess the indications for which the medications were prescribed, which may also influence outcome.

Secondly, the follow-up period was very diverse among patients, (i.e., from 1 to 15 years). Hence, based on the inclusion criteria, a significant drug exposure could range from 6 months to several years, resulting in difficulty interpreting the actual influence of drug exposure among various patients. Thirdly, naturally due to the inconsistent nature of NSAID exposure, we had to define a certain cutoff level. However, as discussed above, the timing of exposure to this class of drugs during disease progression could prove important. Since this is an in vitro study, all drawn conclusions should be cautiously considered.

IPF fibroblasts modify the ECM differently than normal fibroblasts, thus creating a CM that further propagates the ‘IPF-like’ phenotype of normal fibroblasts. We found that PGE2 and Roflumilast inhibit the pro-fibrotic effects of IPF-CM. Further research is needed in order to establish this as a possible treatment for IPF.

## 4. Materials and Methods

### 4.1. IPF-CM Model

Primary HLFs were isolated from IPF tissues and control samples at the time of biopsy, as described [17,41]. Following extraction, HLFs were cultured in DMEM supplemented with 20% FCS, L-glutamine (2 mM) with antibiotics (IMBH, Beit Haemek, Israel) and maintained in 5% CO_2_ at 37 °C. Experiments were performed as previously described [15]. Briefly, IPF/N-HLFs were cultured on Matrigel (Corning, NY, USA). Following 48 h, cells were removed by NH_4_OH, and N-HLFs were added for further culture. PGE2 (1 nM, PeproTech, Cranbury, NJ, USA) and Roflumilast (1 µM, Boringher Ingelheim, Ingelheim am Rhein, Germany) were diluted in DMSO and added an hour prior to the addition of N-HLFs to the culture.

### 4.2. Aggregate Size Measurement

Cell cultures were inspected utilizing an Olympus IX71 microscope. The aggregate size and number were evaluated utilizing the ImageJ software: https://imagej.nih.gov/ij/download.html, accessed on 1 January 2023. Results were normalized to control (N-CM).

### 4.3. Real-Time Quantitative PCR

RNA was extracted by a RNeasy kit (Qiagen, Germany) and converted to cDNA using GeneAmp (Applied Biosystems, Weierstadt, Germany). Reactions were performed with SYBR Green (Applied Biosystems, Carlsbad, CA, USA) according to the manufacturer’s guidelines. Reactions were set to 40 cycles. Primers sequences (purchased from Hylabs, Rehovot, Israel) (5′-3′): GAPDH: F-CTCTGCTCCTCCTGTTCGAC, R-TTAAAAGCAGCCCTGGTGAC; MMP2: F-CAAGGACCGGTTTATTTGGC, R-ATTCCCTGCGAAGAACACAGC; IL8: F-CTCTTGGCAGCCTTCCTGATTT, R-TGGGGTGGAAAGGTTTGGAGTA; COL1A: F-CGAAGACATCCCACCAATCAC, R-CAGATCACGTCATCGCACAAC; ACTA2: F-TGAGAAGAGTTACGAGTTGCCTGAT, R-GCAGACTCCATCCCGATGAA; TGFB1: F-TTTTGATGTCACCGGAGTTG, R-AACCCGTTGATGTCCACTTG. GAPDH was used as control.

### 4.4. Cell Migration

HLFs (5 × 10^4^) were placed in 96-well plates and allowed to adhere for 24 h. Wound closure was monitored immediately after scratching and at 24 h. Areas were measured using ImageJ (http://rsbweb.nih.gov/ij/, accessed on 1 January 2023).

### 4.5. Cell Death

Assessment of apoptosis/necrosis was conducted with AnnexinV-FITC supplemented with PI (MEBCYTO^®^, MBL international, Woburn, MA, USA) by flow cytometry according to the manufacturer’s instructions. AnnexinV+/PI− cells were considered apoptotic, and AnnexinV+/PI+ cells were considered late apoptotic/necrotic. Total cell death was the sum of both. All results are expressed as the percent of the total cell number.

### 4.6. Western Blot

Western blot was performed as previously described [17,41]. Antibodies are listed in Appendix A. Bound antibodies were visualized using goat peroxidase-conjugated secondary antibodies (Appendix A, doi:10.6084/m9.figshare.23708280) followed by enhanced chemiluminescence detection (Millipore, Temecula, CA, USA). LAS3000 (Fujifilm, Tokyo, Japan) was used to quantify protein expressions.

### 4.7. Electronic Record Retrospective Study Population

The study group consisted of a cohort of individuals aged > 18 with IPF according to the guidelines [59].

### 4.8. Data Collection

Data were retrospectively retrieved from patients’ electronical medical records from 5/2003 to 7/2018. Medical records were reviewed and data on serial lung function tests, HRCT of the chest, right heart catheterization and echocardiography were collected for all patients. None of the subjects had a history of relevant occupational or environmental exposure or clinical features of hypersensitivity pneumonitis or connective tissue disease. All patients had negative autoimmune serologic testing.

### 4.9. Statistical Analysis

Statistical analysis was done using GraphPad Prism version 8.00 for Windows (GraphPad Software, La Jolla, CA, USA) and by SPSS (IBM, Armonk, NY, USA). ANOVA was performed to compare differences between multiple cohorts. Paired Student’s *t*-tests were employed to analyze differences between two groups. An effect was considered significant when the *p*-value was <0.05. All experiments were repeated at least three times.

## Figures and Tables

**Figure 1 ijms-24-12393-f001:**
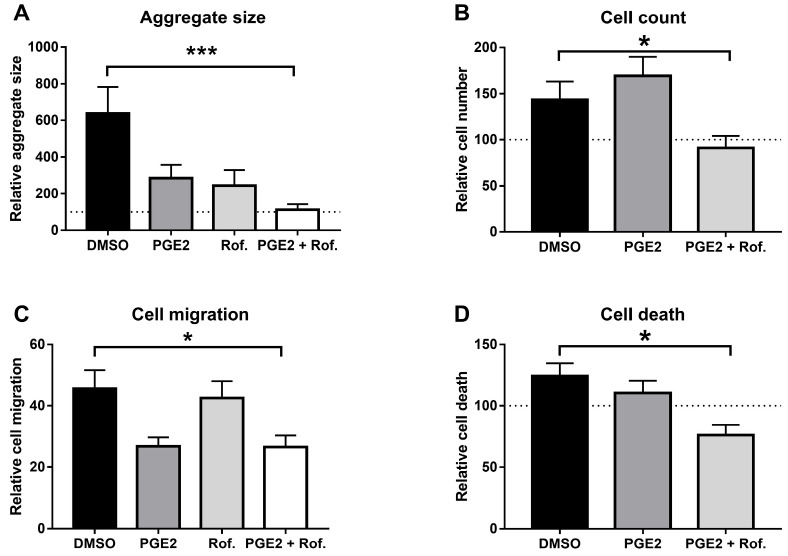
PGE2 and Roflumilast combination inhibits IPF-CMs large aggregate formation. Normal human lung fibroblasts (N-HLF) were cultured on N/IPF-CM for 24 h with/without PGE2 (1 nM) with/without Roflumilast (1 µM) or DMSO. Following culture, relative aggregate size was calculated (**A**), cells were harvested and counted (**B**) and then subjected to cell death analysis using flow-cytometry (**D**). To assess cell migration, N-HLFs were exposed to DMSO and PGE2 (1 nM) with/without Roflumilast (1 µM) by wound-healing assay (**C**,**E**). The results are presented as the relative aggregate size between IPF-CM and N-CM for each treatment with regard to the 100% dashed line, representing zero effect. PGE2—prostaglandin E2, IPF-CM—IPF-conditioned matrix, Rof.—Roflumilast. N ≥ 9, (* *p* < 0.05, *** *p* < 0.0001).

**Figure 2 ijms-24-12393-f002:**
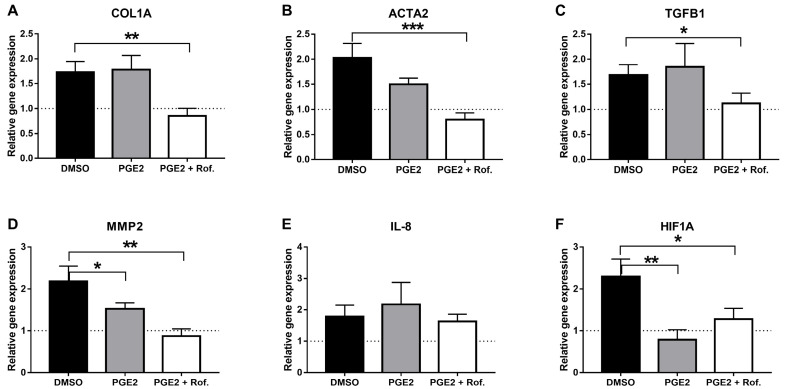
PGE2 and Roflumilast combination inhibits IPF-CMs’ induction of pro-fibrotic gene expression. Normal human lung fibroblasts (N-HLF) were cultured on N/IPF-CM for 24 h with/without PGE2 (1 nM) with/without Roflumilast (1 µM) or DMSO. Following culture, cells were harvested for RNA extraction. The results are presented as the relative gene expression between IPF-CM and N-CM for each treatment, with regard to the 100% dashed line, representing zero effect. (**A**) Collagen type 1 alpha (COL1A), (**B**) alpha-SMA actin alpha 2 (ACTA2), (**C**) Transforming Growth Factor Beta 1 (TGFB1), (**D**) Matrix Metallopeptidase 2 (MMP2), (**E**) Interleukin 8 (IL-8), (**F**) hypoxia-inducible factor 1-alpha (HIF1A). PGE2—prostaglandin E2, IPF-CM—IPF-conditioned matrix. N ≥ 5 (* *p* < 0.05, ** *p* < 0.001, *** *p* < 0.0001).

**Figure 3 ijms-24-12393-f003:**
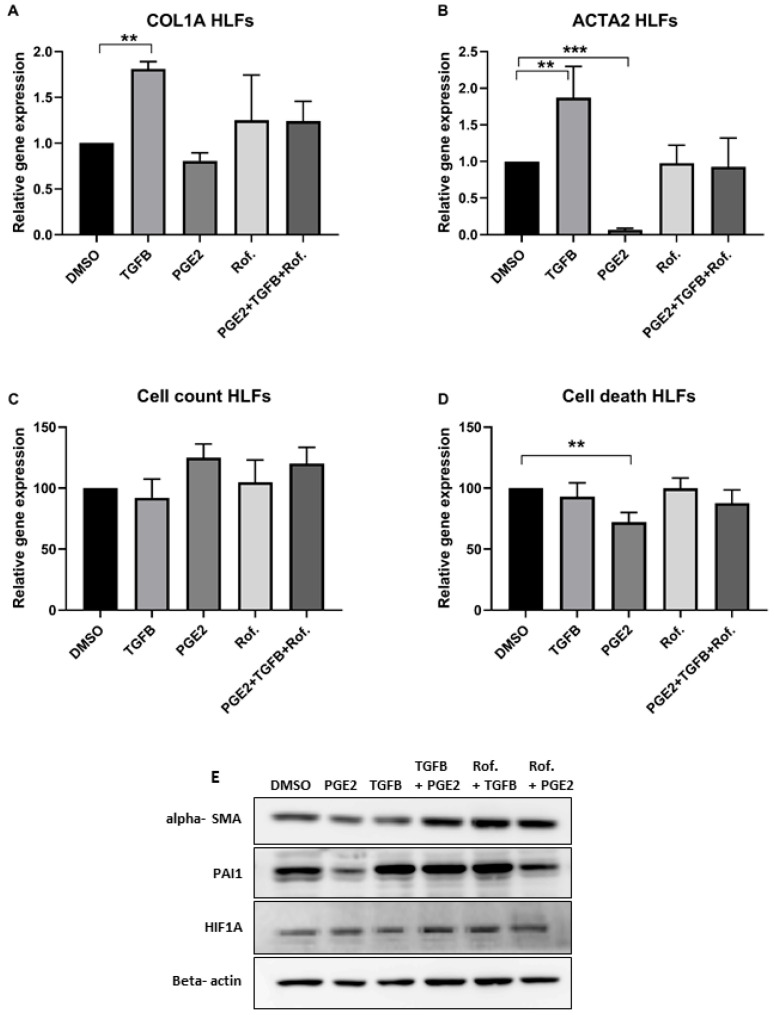
PGE2, TGFβ and Roflumilasts’ direct effect of HLFs following 24 h. Normal human lung fibroblasts (N-HLF) were cultured for 24 h with/without PGE2 (1 nM), TGFβ with/without Roflumilast (1 µM) or DMSO. Following culture, cells were harvested for RNA/protein extraction (**A**,**B**,**E**) as well as counted (**C**) and analyzed for cell death using flow cytometry (**D**). The results are normalized to control DMSO. HIF1A—hypoxia-inducible factor 1-alpha. N ≥ 3. (** *p* < 0.001, *** *p* < 0.0001).

**Figure 4 ijms-24-12393-f004:**
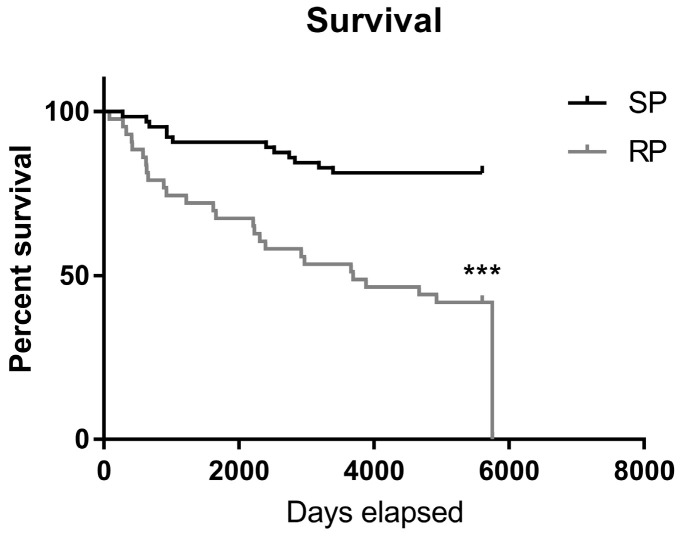
Survival curve of patients from SP and RP groups. Patient survival was analyzed using the Kaplan–Meier plot (*** indicates *p* < 0.001).

**Figure 5 ijms-24-12393-f005:**
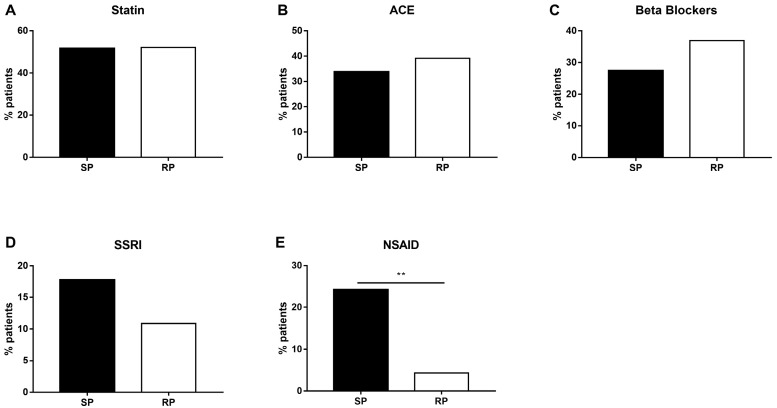
Medication exposure in IPF patients during the study period. Medication exposure to the following five classes of drugs: statins (**A**), angiotensin-converting enzyme (ACE) inhibitors/angiotensin type 1 receptor blockers (ARB1) (**B**), beta blockers (**C**), serotonin-specific reuptake inhibitors (SSRI) (**D**) and NSAIDs (including COX-2 inhibitors) (**E**) were examined, starting 6 months prior to diagnosis of IPF until end of follow-up. Comparison was made between the slow and rapid progressors (SP vs. RP). (** indicates *p* < 0.01).

**Table 1 ijms-24-12393-t001:** Patient characteristics according to the slow/rapid progressing IPF.

Parameter	Slow Progression N = 62	Rapid Progression N = 45	*p*-Value
Age, years	65.7 ± 12.3	66.3 ± 10.3	0.39
Gender (% male)	56%	67%	0.02
Smoking history	77%	78%	0.79
Cardiovascular disease	27%	24%	0.41
Renal failure	2%	9%	<0.001
Malignancy	25%	0	<0.001
Mortality	19%	54%	<0.001

**Table 2 ijms-24-12393-t002:** Pulmonary function tests of patients according to the slow/rapid progressing IPF.

Parameter	Slow Progression N = 62	Rapid Progression N = 45	*p*-Value
FVC, L	2.34 ± 0.82	1.85 ± 0.75	0.002
FVC, % predicted	77.51 ± 21	59.46 ± 17.9	0.001
FRC PL % predicted	79.24 ± 20.1	62.6 ± 15.4	<0.001
FRC, L	2.46 ± 0.66	1.98 ± 0.57	0.001
TLC, L	4.16 ± 1.11	3.22 ± 0.88	<0.001
TLC, % predicted	72.63 ± 16.6	57.23 ± 11.5	<0.001
DLCO, % predicted	54.51 ± 15	42.27 ± 13.2	<0.001
6 min walk test			
SAT rest, %	96.5 ± 2	95.13 ± 2.3	0.017
SAT exercise, %	90.90 ± 5.3	86.17 ± 4.8	0.001
HR rest	78.31 ± 10.3	78.08 ± 12.6	0.940
HR exercise	110.2 ± 18	107.6 ± 20	0.619
Distance, m	384.4 ± 150.4	307 ± 153.9	0.053

Forced vital capacity (FVC), functional residual capacity (FRC), total lung capacity (TLC), single-breath diffusing capacity for carbon monoxide (DLCO), saturation (SAT), heart rate (HR).

## Data Availability

Data will be made available upon request.

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
