# Peer review of "Prostaglandin E2 (PGE2) and Roflumilast Involvement in IPF Progression"

_ijms, 2023, doi:10.3390/ijms241512393_

Round 1
Reviewer 1 Report
The paper is interesting however the evidence of the contribution of PGE2 in IPF has been already widely demonstrated. The synergic effect with Roflumilast is not sufficiently supported by the data. Further mechanistic experiments are required as well as some preclinical models to demonstrate an impact on pathology progression by the pharmacological association that is suggested
Author Response
The paper is interesting however the evidence of the contribution of PGE2 in IPF has been already widely demonstrated. The synergic effect with Roflumilast is not sufficiently supported by the data. Further mechanistic experiments are required as well as some preclinical models to demonstrate an impact on pathology progression by the pharmacological association that is suggested.
Response: We thank the reviewer for his input. Indeed, the benefit of Roflumilast was previously suggested by others, using mouse BLM models. This information was added to the introduction and the discussion section. Due to the growing understanding that mouse physiology isn’t always correlated to humans, especially in the case of IPF (e.g. lacking the UIP pattern, spontaneous recovery etc.), it is believed additional experimental models are needed. Our system shows results in a human derived primary cell culture 3D model. Our experimental system was previously used to show benefits of Nintedanib and Pirfenidone [15-17], thus supporting previous results in an additional experimental ex-vivo system. Unfortunately, we cannot provide further preclinical models.
Reviewer 2 Report
The authors described an interesting combination of PGE2 with roflumilast in delaying the process of fibrogenesis. The results are showing that the combination treatment of PGE2 and roflumilast lower down the expression level of ACTA2, COL1A1, TGFB1, and MMP2. It also inhibited TGFb induced overexpression of COL1A1, CTA2, and PAI1. Results are convincible and supportive for the conclusion that PGE2 with roflumilast might be one of the potential anti-fibrotic drugs. However, the authors also described the data of how patient used different drugs during slow or fast fibrogenesis progression. Please explain the biological correlation of PGE2+roflumilast treatment and data present in slow/fast fibrogenesis as well as the drug selections among patient. Is there any mouse data available to support that PGE2+roflumilast can be one of the potential drugs?
In figure 1, it showed the PGE2+ roflumilast inhibited the cell growth as well as cell viability, the inhibition of fibroblast differentiation markers might be due to the lower viability, which has no direct correlation in inhibiting the dysregulated repair mediated by fibroblast during fibrogenesis.
One last thing, please add more details in method section, for example: aggreation measurement, qRT-PCR cycle details, cell death (please described the details). and there is no western blot method section.
Author Response
The authors described an interesting combination of PGE2 with roflumilast in delaying the process of fibrogenesis. The results are showing that the combination treatment of PGE2 and roflumilast lower down the expression level of ACTA2, COL1A1, TGFB1, and MMP2. It also inhibited TGFb induced overexpression of COL1A1, CTA2, and PAI1. Results are convincible and supportive for the conclusion that PGE2 with roflumilast might be one of the potential anti-fibrotic drugs. However, the authors also described the data of how patient used different drugs during slow or fast fibrogenesis progression.
- Please explain the biological correlation of PGE2+roflumilast treatment and data present in slow/fast fibrogenesis as well as the drug selections among patient.
Response: The drugs selected in the last section refer to the most abundant treatments given routinely to the general population (not including specific disease related treatments such as DM etc.). From our experience, these include mostly anti-depressants (SSRI), pain/ inflammation (NSAIDs) and treatments for high blood pressure/ heart condition (e.g. beta blockers, ACE inhibitors). We agree that there could be further granularity for this analysis. However, given the relatively low numbers of patients included, it was decided to combine treatment regimens in order to achieve significance. Moreover, we sought to include IPF patients only, without COPD, and therefore, unfortunately, no Roflumilast was not present in the prescribed treatments. The reasons for selecting these treatment groups were added to the discussion section, as well as to the results section. Moreover, an explanation regarding the correlation was added to the discussion section.
- Is there any mouse data available to support that PGE2+roflumilast can be one of the potential drugs?
Response: Unfortunately no mouse data is available. Evidence from experiments with PDE4 inhibition in mouse BLM models were added to the discussion section.
- In figure 1, it showed the PGE2+ roflumilast inhibited the cell growth as well as cell viability, the inhibition of fibroblast differentiation markers might be due to the lower viability, which has no direct correlation in inhibiting the dysregulated repair mediated by fibroblast during fibrogenesis.
Response: We agree that cell viability has no direct correlation in inhibiting the dysregulated repair mediated by fibroblast during fibrogenesis. However, we felt that it is important to describe the cell phenotype (e.g. cell count, migration etc.) as the possible reasons for the increased aggregate size. The data in figure 1 shows the relative effect of the IPF-CM vs. the normal-CM. This data relies on numerous experiments performed using this system, showing that the IPF-CM by itself increases HLF cell counts, cell migration and pro-fibrotic gene expression [15-17]. Here, we exposed the cells to either PGE2 or the PGE2+Roflumilast combination and observed the change in the effect of the IPF-CM. This method was previously used to describe the effects of nintedanib and pirfenidone in this system. The effect on cell count is shown in figure 1B, showing that the combination prevented the elevation in cell counts (i.e. similar numbers of cells are found in IPF-CM and N-CM following exposure to the combination), thus no reduced viability was indicated. In figure 1D, we show that HLFs cultured on IPF-CM usually have increased cell death of about 20% increase. Again, the combination of PGE2+ Roflumilast prevented this modest increase in cell death, thus indicating on improved cell viability. From these results, and our previous experience using this system, we can carefully assume the effect on gene expression to be direct, without the bias of reduced cell viability. To support this assumption, we conducted another experiment in which we tested cell death on HLFs cultured in the presence of the treatments described in Figure 3. These results were added to Figure 3, in addition to cell counts (now Figure 3C+3D).
- One last thing, please add more details in method section, for example: aggreation measurement, qRT-PCR cycle details, cell death (please described the details). and there is no western blot method section
Response: Thank you for this comment. The methods section was revised accordingly.
Round 2
Reviewer 2 Report
Thanks for the hard work from authors for answering most of my comments. However, I am still confused the biological relations of the clinical drug selection for fibrosis treatment and PGE2 or Roflumilast in inhibiting the fibrogenesis. The function of PGE2 can be either proinflammatory lipid mediator or anti-inflammatory agents. Must be clear of the role of PGE2 and Roflumilast in fibrosis treatment.
Author Response
Thanks for the hard work from authors for answering most of my comments.
- However, I am still confused the biological relations of the clinical drug selection for fibrosis treatment and PGE2 or Roflumilast in inhibiting the fibrogenesis.
Response: Thank you for this review. We extended the section discussing the concomitant medications, in order to clarify the logic behind the selection of the treatment groups, their possible connection to IPF progression and further elaborating about the results. Additional references were added to support the experimental design (page 8).
- The function of PGE2 can be either proinflammatory lipid mediator or anti-inflammatory agents. Must be clear of the role of PGE2 and Roflumilast in fibrosis treatment.
Response: Following this comment, we added additional data in the discussion section explaining the possible function and mode of action of the PGE2 alone and PGE2 + Roflumilast combination in our system. We added additional references that support our data in other experimental models (page 7).